# Hexokinase 2 in Cancer: A Prima Donna Playing Multiple Characters

**DOI:** 10.3390/ijms22094716

**Published:** 2021-04-29

**Authors:** Francesco Ciscato, Lavinia Ferrone, Ionica Masgras, Claudio Laquatra, Andrea Rasola

**Affiliations:** 1Dipartimento di Scienze Biomediche, Università di Padova, 35131 Padova, Italy; lavinia.fer@gmail.com (L.F.); ionica.masgras@gmail.com (I.M.); claudiolaquatra@gmail.com (C.L.); 2Institute of Neuroscience, National Research Council, 56124 Pias, Italy

**Keywords:** hexokinase 2, mitochondria, MAMs, Ca^2+^, tumor metabolism, apoptosis, chemotherapeutics, cell-penetrating peptides

## Abstract

Hexokinases are a family of ubiquitous exose-phosphorylating enzymes that prime glucose for intracellular utilization. Hexokinase 2 (HK2) is the most active isozyme of the family, mainly expressed in insulin-sensitive tissues. HK2 induction in most neoplastic cells contributes to their metabolic rewiring towards aerobic glycolysis, and its genetic ablation inhibits malignant growth in mouse models. HK2 can dock to mitochondria, where it performs additional functions in autophagy regulation and cell death inhibition that are independent of its enzymatic activity. The recent definition of HK2 localization to contact points between mitochondria and endoplasmic reticulum called Mitochondria Associated Membranes (MAMs) has unveiled a novel HK2 role in regulating intracellular Ca^2+^ fluxes. Here, we propose that HK2 localization in MAMs of tumor cells is key in sustaining neoplastic progression, as it acts as an intersection node between metabolic and survival pathways. Disrupting these functions by targeting HK2 subcellular localization can constitute a promising anti-tumor strategy.

## 1. Introduction

Hexokinases (HKs) are evolutionarily conserved enzymes that phosphorylate six-carbon sugars (hexoses). Phosphorylated glucose (glucose-6-phosphate) is metabolized in glycolytic, glycogenic, pentose phosphate and hexosamine biosynthesis pathways, playing key roles in ATP synthesis, glucose storage, NADH pool enrichment and protein glycosylation. In mammals, four HK isozymes (HK1-4) have been characterized [1], and a fifth isozyme named HKDC1 (HexoKinase Domain Containing protein 1) has been more recently described [2]. HK1-3 and HKDC1 are 100 kDa proteins that probably result from the duplication of an ancestral HK gene similar to HK4, a 50 kDa enzyme with a single catalytic moiety expressed at high levels in liver and pancreatic β-cells, where it controls glycogenesis and insulin secretion, respectively. In contrast to HK4, all other HKs have two enzymatic pockets, but these are both functional only in HK2, whereas HK1, HK3 and HKDC1 retain activity only in the C-terminal catalytic site. HK1 is ubiquitous and constitutes the predominant HK form in most tissues; HK2 is broadly present but less expressed than HK1, and it is the main isozyme in insulin-sensitive districts such as heart, skeletal muscle and adipose tissue, and in a wide range of tumors; HK3 and HKDC1 have a more limited pattern of expression and have been less characterized [3,4]. All HKs are cytosolic, but HK1 and HK2 can also dock to mitochondria through a N-terminal motif absent in the other isoforms. When bound to mitochondria, HK1 and HK2 exert cytoprotective effects in healthy and neoplastic cells and increase their efficiency in glucose usage [5,6].

## 2. HK2 Regulation: Where, How and Why

HK2 levels and activity are dynamically regulated by many factors that act at transcriptional, translational and post-translational levels (Figure 1). In general, HK2 confers protection from stress states and contributes to proper tissue development. It is induced by growth factors such as FGF [7] or TGF-β [8] that impinge upon activation of the transcription factor c-MYC, of which the HK2 gene is a target [9]. FGF signaling is important for vascularization, and HK2 ablation in endothelial and lymphatic cells leads to blood and lymphatic vascular defects mirroring FGF receptor (FGFR)-KO mice [7]. In fibroblasts, HK2 enhances the pro-fibrotic effect of TGF-β, whereas HK2 knock-down decreases the TGF-β-stimulated fibrogenic process [8]. Ectopic overexpression of the HK2 gene in brain substantia nigra protects dopaminergic neurons from degeneration in acute mouse models of Parkinson’s disease [10]. In cardiomyocytes, HK2 protects from H_2_O_2_ injury and prevents heart maladaptive hypertrophy increasing the pentose phosphate pathway (PPP) flux [11]. Conversely, HK2^+/−^ mice are vulnerable to ischemia/reperfusion cardiac damage [12] and are affected by ROS-mediated cardiac hypertrophy [13]. However, heterozygous animals have a proper development without defects in insulin sensitive tissues [14]. Genetic models are poorly informative on HK2 functions, as HK2 knock-out mice undergo embryonic lethality [14] and knocking-out HK2 in adult mice does not cause evident defects [15].

HK2 induction is part of the anabolic program elicited by insulin in muscle tissues (Figure 1). Insulin prompts HK2 expression following activation of PI3K and p70 S6 protein kinase (p70S6K) downstream to mTOR complex 1 (mTORC1) [16,17], a central coordinator of cellular biosynthetic programs [18]. Consistently, HK2 expression is decreased in type I and type II diabetes [4,14]. In addition, activation of the mTORC2 complex elicits HK2 mitochondrial docking in insulin-treated cells [19].

HK2 activity and localization are dynamically regulated by its phosphorylation (Figure 1). In cardiomyocytes, mitochondrial HK2 displays an anti-apoptotic function mediated by AKT [20], a serine/threonine kinase that acts downstream to a variety of growth factors as a central anabolic and survival effector [21]. AKT enables HK2 docking to mitochondria by directly phosphorylating its Thr-473 residue [22] and by inhibiting glycogen synthase kinase-3β (GSK-3β) activity that prevents HK2 mitochondrial localization [23]. While the phosphatases that target the HK2 Thr-473 amino acid are still unknown, the PH domain leucine-rich repeat protein phosphatase (PHLPP) inhibits AKT, reducing the levels of HK2 bound to mitochondria in heart [24]. When on the outer mitochondrial membrane (OMM), HK2 exerts an anti-apoptotic function by inhibiting the mitochondrial Permeability Transition Pore (mPTP) [25], a mitochondrial channel the opening of which triggers cell death [26,27]. Under oxidative stress conditions, HK2 association to mitochondria in rhabdomyosarcoma cells is regulated by Myotonic Dystrophy Protein Kinase (DMPK) and Src kinase in a multi-molecular complex characterized by antioxidant and pro-survival properties involved in muscle fiber differentiation [28].

Mitochondrial binding is crucial for the protective effects of HK2 in heart. Knocking-out PHLPP increases AKT activity and decreases the infarcted area after ischemia/reperfusion damage [24]. Several cardioprotective interventions such as ischemic preconditioning, morphine and insulin increase the mitochondrial binding of HK2, whereas its disruption abrogates the protective potential of ischemic preconditioning [29]. More in general, mitochondrial HK2 shields from cardiac ischemia/reperfusion injury [22], at least partially by slowing adenine nucleotide exchange via inhibition of Voltage Dependent Anion Channel (VDAC) on the OMM [30,31]. In further accord with an important survival role of mitochondrial HK2, the AKT/HK2 axis shields neurons from ischemia [32,33].

In case of nutrient paucity, cells can activate autophagy through mTORC1 inhibition [18] in order to guarantee cellular homeostasis and survival [34]. HK2 contributes to autophagy induction in conditions of glucose starvation by binding and inhibiting mTORC1. As a result, mTORC1 inhibition leads to activation of ULK1, a kinase that triggers the first steps of the autophagic process [4,35]. HK2 exerts its pro-autophagic action by binding the Raptor subunit of mTORC1 through a TOS (mTOR signaling) motif unrelated to the HK catalytic site and to the mitochondrial binding sequence and not present in the other HK isoforms [35]. HK2 involvement in autophagy induction in heart and tumors and in hypoxic conditions was further confirmed and characterized [36,37,38]. Moreover, HK2-induced autophagy was proposed to be involved in chemotherapeutic resistance of some tumor models [39,40].

HK2-dependent inhibition of mTORC1 leads to inactivation of the mTORC1 substrate p70S6K [35], a key stimulator of protein synthesis and cell growth [41] also involved in the anabolic induction of HK2 [16,17]. Hence, p70S6K inhibition acts as a feedback inhibitory mechanism to block HK2 induction in case of nutrient shortage, with a potential relevance in the fluctuating conditions of metabolic supply occurring during the process of neoplastic growth.

Little is known about HK2 functions in the immune system. It was found that HK2 can act as an innate immune receptor for bacterial-derived N-AcetylGlucosamine (NAG) in macrophages. NAG binding to HK2 elicits its detachment from mitochondria and leads to NLRP3 inflammasome activation and to the ensuing pro-inflammatory IL-1β secretion [42]. This unprecedented HK2 function highlights the diversity of the biological processes in which the enzyme can be involved.

In an in vitro model, HK2 elicits monocyte differentiation through induction of autophagy, and this effect is antagonized by the sirtuin SIRT6 [43]. As macrophage differentiation requires a complex metabolic reprogramming, with an induction of glycolysis that characterizes the pro-inflammatory and anti-tumoral M1 type macrophages [44], a wider role of HK2 in tuning this process can be envisaged and needs further investigation.

Furthermore, during the process of hepatocarcinogenesis hexokinase expression can switch from isozyme HK4 to HK2, making hepatocellular carcinoma (HCC) cells insensitive to the anti-neoplastic activity of Natural Killer (NK) cells [45], even if the mechanisms underlying this regulation of innate immunity remain to be elucidated.

A recent study shows that genetic ablation of HK2 in mouse T cells does not affect their activation and in vivo viral immunity, and that HK2 is not required for Treg function. This suggests that HK2 can constitute a target for antineoplastic therapy, as its inhibition would not interfere with T cell activity [46].

Interestingly, displacement of HK2 from mitochondria of dendritic cells was proposed to be involved in their death during SARS-CoV2 infection [47].

## 3. HK2 on Mitochondria of Neoplastic Cells: Maintaining the Bad Guy in Perfect Shape

During neoplastic growth, tumor cells need to sustain a deregulated cell replication even in a harsh metabolic milieu by adjusting their biochemical responses to rapid fluctuations in oxygen and nutrient availability [48]. In this scenario, mitochondrial HK2 intersects both metabolic and survival pathways, thus allowing neoplastic cells to cope with stressful environmental cues. In accord with its relevance in tumor settings, HK2 de novo expression or overexpression is related to poor prognosis, stage progression, metastasis and/or treatment resistance in a variety of malignancies [49]. As an example, during the evolution of HCC, HK2 expression replaces glucokinase (HK4) in hepatocytes, concomitantly with their metabolic switching towards aerobic glycolysis [50]. HK2 expression already occurs in precancerous lesions [51] and may be considered as a predictive biomarker for HCC development [52]. HCC treatment with thyroid hormone T3 in a rat model of hepatocarcinogenesis switches back to HK4 expression and is associated with a metabolic reprogramming towards an oxidative phosphorylation (OXPHOS) phenotype, abrogating HCC progression [53]. In addition, HK2 is connected to poor prognosis (Table 1) in colorectal cancer [54], glioblastoma multiforme [55], pancreatic cancer [56], cervical squamous cell carcinoma [39,57], prostate cancer [58], head and neck squamous cell carcinoma [59], HCC [45] and other tumor types [3], and two independent meta-analyses in tumors of the digestive system indicate HK2 as a negative prognostic marker associated with shorter overall survival and progression free survival [60,61]. (Over)expression of HK2 is also linked to tumor progression and/or metastasis in a variety of diverse neoplastic models (Table 1) including HCC [62], colorectal carcinoma [54], lung cancer, breast cancer [15], prostate cancer [58,63,64], diffuse large B-cell lymphoma [65] and glioblastoma [66].

HK2 induction associates with resistance to radiation therapy [57,66], to chemotherapeutics (Table 1) including the pyrimidine analog gemcitabine [56], the DNA alkylating compound cisplatin [40], the microtubule stabilizing agent paclitaxel [38], estrogen receptor modulation with 4-hydroxytamoxifen [39], and to targeted therapies such as the kinase inhibitor Sorafenib [67] and the anti CD20 antibody Rituximab alone or in combination with cyclophosphamide, doxorubicin, vincristine and prednisone (the R-CHOP regimen) [68].

Noteworthy, HK2 pharmacological inhibition or genetic downregulation suppress tumor growth or reactivate therapeutic sensitivity in a number of tumor cell models [3]. For instance, knocking-out HK2 expression prevents or strongly inhibits tumor formation in genetic mouse models of lung and breast cancer [15] and of PTEN-negative prostate cancer [63], while liver-specific ablation of HK2 decreases tumor incidence in a mouse model of hepatocarcinogenesis [62]. Altogether, these observations point towards a mechanistic connection between HK2 expression and neoplastic growth. The functions of HK2 in tumor cells encompass metabolic rewiring towards aerobic glycolysis, tuning of autophagy for handling nutrient shortages and shielding from cell death stimuli [4]. Overall, this broad range of activities can presumably concur in explaining HK2 remarkable association with a multiplicity of tumor types and with their elevated malignancy, as well as with the resistance to diverse antineoplastic treatments.

It is therefore of paramount importance to finely dissect HK2 regulation, subcellular distribution and biochemical functions in tumor models, with the ultimate goal of identifying a selective and efficient mode of targeting it.

HK2 induction in cancer is strictly intertwined with several pathways and factors (Figure 2), the deregulation of which drives neoplastic progression. When tumor cells must cope with hypoxic conditions or with pseudohypoxia elicited by a rise in oncometabolite levels [69], stabilization of HIF1α and the consequent orchestration of the HIF1-dependent transcriptional program confer them increased motility, while favoring angiogenesis, decreasing OXPHOS and enhancing glucose usage [70]. HK2 is a transcriptional target of HIF1 that contributes to this metabolic rewiring under (pseudo)hypoxic conditions [50,71]. Oncogenic hyperactivation of key proto-oncogenes, including RAS, c-MYC or ERBB2, increases HK2 levels in several neoplastic models [7,15], and induction of the RELA/p65 transcription factor drives HK2 and lymphomagenesis in the central nervous system [72]. Loss of the tumor suppressors PTEN and p53 induces HK2 in prostate cancer [64]. In a model of KRAS-driven lung cancer, antioxidants stabilize the transcription factor BACH1, enhancing HK2 transcription and triggering a metastatic spread favored by glycolytic metabolism of cancer cells [73]. It is possible to envisage a positive feedback loop in metastatic cells where the KRAS-induced HK2-expression decreases oxidative stress [74], possibly by increasing the activity of the PPP, stabilizes BACH1 and further enhances HK2 expression.

HK2 levels are also post-transcriptionally regulated by miRNAs. HK2 expression is enhanced by pro-tumorigenic miR-155 that decreases the tumor suppressor miR-143, which targets HK2 mRNA and reduces its protein levels [75]. Other miRNAs, such as miR-216a-5p, miR-125a and miR-185 down-regulate HK2 levels and aerobic glycolysis in models of HCC, osteosarcoma and melanoma [76,77,78].

In several tumor types AKT is constitutively active [21], leading to HK2 phosphorylation on Thr-473 and to its localization on the OMM [22], where it inhibits cell death blocking mPTP opening [25,79]. It was recently shown that also a second kinase, PIM2, can phosphorylate the same HK2 Thr-473 residue [38]. The importance of this post-translational modification is highlighted by the observation that a non-phosphorylatable HK2^T473A^ mutant decreases tumorigenesis and metastasis in colon and breast cancer xenograft models [80].

HK2 localization to mitochondrial surface is further promoted through the ubiquitination of the Lys-63 residue by HECTH9 E3 ligase, which elicits HK2 binding to VDAC on the OMM and the consequent expansion, metabolic rewiring and chemoresistance of cancer stem cells in a prostate cancer model [58]. Both Thr-473 and Lys-63 are absent in the other HK isozymes, suggesting that HK2 binding to mitochondria underpins unique and distinct biochemical functions. In addition, it was recently demonstrated that HK2 can be SUMOylated at Lys-315 and Lys-492 residues and that de-SUMOylated HK2 preferably binds to mitochondria [81].

The AKT/HK2 signaling pathway is tuned by PHLPP, the only known phosphatase to form a complex with AKT and HK2 on the mitochondrial surface. PHLPP decreases HK2 phosphorylation, inducing its translocation from mitochondria to the cytosol, and reverts the glycolytic phenotype of colon cancer cells, repressing their proliferative potential [82]. PHLPP switches off AKT through its dephosphorylation, and it is unclear whether it can also directly act on the Thr-473 residue of HK2 [82]. Interestingly, PHLPP is induced by mTORC1 and in turn inactivates the mTORC1 substrate p70S6K [83]. This scenario opens the possibility of complex regulatory networks that tune HK2 levels and subcellular localization in order to match them with the metabolic status of the cell. Indeed, HK2 is induced by the mTORC1/p70S6K axis, which can instead inhibit HK2 under conditions of glucose paucity, when AKT is presumably more dephosphorylated by PHLPP. These dynamic and intricate feedback loops could play a role in autophagy regulation and in neoplastic cell survival under stress conditions, and their detailed dissection deserves further investigations.

Under hypoxia HK2 can form a complex on the mitochondrial surface with TIGAR, a fructose-2,6-bisphosphatase that inhibits glycolysis and promotes PPP [84]. The HK2/TIGAR complex lowers mitochondrial ROS levels, stimulates HK2 activity and protects colorectal carcinoma cells from death. [85]. Hence, this functional interaction is relevant in maintaining redox homeostasis, even though the underlying molecular mechanisms for mitochondrial ROS regulation are still elusive.

Similarly, how mitochondrial HK2 antagonizes death stimuli remains poorly understood. HK2 inhibits the activity of pro-apoptotic proteins of the Bcl-2 family [4], even if it can protect cells from death also independently of them [86], suggesting the presence of multiple mechanisms of action. HK2 displacement from mitochondria triggers cell death through mPTP opening [25]. However, the functional interplay between HK2 and mPTP must be indirect, as the enzyme binds to the external surface of the OMM through a hydrophobic sequence of 15 amino acids at its N-terminal domain [87], whereas the mPTP is in the inner mitochondrial membrane, where it is probably constituted by the F-ATP synthase [26]. Increased ROS levels and Ca^2+^ accumulation in the mitochondrial matrix are among the factors that elicit mPTP opening and could be kept under control by the mitochondrial HK2 fraction.

Disruption in HK2 binding to the OMM with small molecules or selective peptides leads to tumor cell death or sensitization to chemotherapeutic agents [25,86,88,89], making a precise definition of HK2 docking site(s) on mitochondria mandatory to understand how it shields tumor cells from noxious stimuli. Several experimental evidences indicate that HK2 interacts with VDAC1 [5,90], and HK2 binding to VDAC1 lowers the probability of detrimental mPTP opening and consequent cell death under stress conditions [90,91]. However, HK2 detachment from mitochondria elicits apoptosis also in double VDAC1/3^−/−^ knock-out cells [25], implying that other HK2 interactors on mitochondria exist and contribute to its pro-survival function.

A further characterization of HK2 binding to mitochondria came with the recent demonstration [92] that it lodges in domains called Mitochondria-Associated Membranes (MAMs; Figure 3), small subcellular compartments where the OMM is in close contact with the endoplasmic reticulum (ER), with the two membranes separated by a space of only 10 to 50 nm (tight and loose MAMs, respectively). MAMs are involved in several metabolic and signaling pathways, encompassing glucose and lipid homeostasis, ER stress, autophagy and cell death, and they tune Ca^2+^ signaling to mitochondria [93]. Remarkably, up to 80% of HK2 localizes in MAMs of diverse neoplastic cell types, including cervix, colon and breast carcinoma models and cells derived from neurofibromas and from malignant peripheral nerve sheath tumors, whereas only about 30% of HK2 colocalizes with MAMs in non-tumor cells such as macrophages and myoblasts [92]. It is therefore possible to envision that MAM localization of HK2 could contribute to specific processes required for the fitness of cancer cells, such as the tight maintenance of intracellular and mitochondrial Ca^2+^ homeostasis. Mitochondria require basal Ca^2+^ uptake for enzymatic activities and survival [94], and cancer cells are particularly vulnerable to any interference to Ca^2+^ transfer to mitochondria, as they rapidly undergo a lethal bioenergetic catastrophe [95]. This indicates that neoplastic cells are endowed with a strict dependence from basal Ca^2+^ supply to mitochondria for their metabolic functions, such as the optimal activity of dehydrogenases, whereas at the opposite side a rapid Ca^2+^ surge in the matrix can prompt mPTP opening and cell death [96].

Ca^2+^ transfer from ER to mitochondria requires the activation of Ca^2+^ channels in the ER membrane called IP_3_ receptors (IP_3_Rs), which are largely localized in loose MAMs [97], i.e., in close proximity to HK2 [92]. Ca^2+^ released from ER through IP_3_Rs is taken up by the mitochondrial Ca^2+^ uniporter (MCU), and its flux across MAMs is regulated by a multimeric complex including IP_3_Rs on the ER side, VDAC in the OMM and the connecting chaperone GRP75. Further layers of control exist in order to avoid an unrestrained Ca^2+^ flux into mitochondria, such as AKT-dependent phosphorylations (for a complete review see [98]). In many tumor cells, HK2 displacement from MAMs rapidly elicits a massive Ca^2+^ flux to mitochondria that leads to mPTP opening and apoptosis and can be blocked by IP3R inhibition [92]. These observations strongly suggest that HK2 is involved in handling ER-mitochondria Ca^2+^ fluxes, and highlight the vital importance of keeping them under control for neoplastic cells. HK2 interacts with the chaperone GRP75 [92] and with AKT, which inhibits IP_3_R opening by directly phosphorylating it in MAMs [98]. All these experimental clues lead to hypothesize that HK2 participates in an IP_3_R-GRP75-VDAC1 complex in MAMs, where it could prevent harmful Ca^2+^ fluxes to mitochondria without affecting physiological Ca^2+^ transfer. An extensive study on the dynamics and regulation of this protein complex is mandatory to unveil its roles in favoring tumor cell adaptations to environmental challenges that mobilize Ca^2+^.

## 4. Spotting and Picking off HK2 in Tumors: A Matter of Precision

The functions carried out by mitochondrial HK2 are required to confer tumor cells the ability to thrive and adapt even under harsh conditions of metabolic fluctuations and exposure to noxious stimuli. Consequently, targeting HK2 constitutes a fascinating perspective for the development of novel anti-neoplastic tools (Figure 3). In the last years, several HK inhibitors have been investigated, including the catalytic inhibitors 3-Bromopyruvate (3-BrPyr) and Lonidamine and the glucose-analogue, competitive inhibitor 2-Deoxyglucose (2-DG). These molecules target HK2 in many in vitro and in vivo tumor models, detach it from mitochondria and elicit tumor cell death [3]. In the case of 3-BrPyr, the best characterized of these molecules, it perturbs the HK2-VDAC interaction on the mitochondrial surface of cancer cells [99] and kill them by apoptosis or necrosis [100]. 3-BrPyr has anti-tumor activity in several in vivo models [56], with a certain degree of selectivity for malignancies, probably explained by the overexpression of its transporter MCT1 in high glycolytic tumors cells [101], and in association with classical antitumor drugs, 3-BrPyr can revert cancer cell chemoresistance [67,102,103]. However, 3-BrPyr can have multiple additional effects caused by its alkylating nature and high reactivity toward a variety of proteins, such as dehydrogenases, whereas its conjugation with reduced glutathione perturbs intracellular redox homeostasis [104]. These non-specific targets could explain the reported adverse effects, including liver and gastrointestinal toxicity in animal models [105] and hepatic and renal toxicity in humans [106], as well as the death of three oncological patients after 3-BrPyr administration by a nonmedical practitioner [104]. Consequently, no clinical trials were approved for 3-BrPyr. 2-DG can sensitize chemoresistant cancer cells to cisplatin or paclitaxel, but its use was abandoned after the onset of adverse effect and toxicity during clinical trials, while Lonidamine turned out to be ineffective in Phase II and III clinical trials [3]. Importantly, these molecules inhibit all HKs without any specificity for HK2, with the evident risk of decreasing or suppressing glucose phosphorylation and utilization in crucial organs such as brain, heart, kidney and liver. To circumvent this problem, experiments are under way with new formulations in order to specifically deliver HK inhibitors to tumors [3]. The antifungal azole derivatives ketoconazole and posaconazole have shown some efficacy in targeting HK2 and inducing apoptosis in cancer cell models [107], and one phase 0 and two early phase 1 clinical trials are starting in glioma patients. However, non-specific effects were reported in tumor cells with the similar compound clotrimazole [23], and at present no HK inhibitor has progressed toward more advanced trial phases, nor it has reached the clinical practice.

An alternative strategy for targeting HK2 in tumor cells is the use of specific peptides that displace HK2 from OMM without affecting the catalytic activity of any HK. These molecular tools carry cell-penetrating sequences in order to cross the plasma membrane [108] and are extremely effective in eliciting a fast and massive death of cells derived from diverse tumor models [25,28,86,92,109]. The death pathway activated by HK2 detachment from MAMs has been shown to rely on a rapid Ca^2+^ release from IP_3_Rs in the ER and on a further Ca^2+^ entry through plasma membrane channels. The ensuing mitochondrial Ca^2+^ overload results in calpain activation, mPTP opening, mitochondrial depolarization and death of all tumor cells in less than one hour [92]. However, these peptides lack specificity for tumor cell entry as cell-penetrating sequences can transport cargoes inside any cell, which makes such molecules not suitable for in vivo utilization.

A possibility to bypass this problem is offered by the use of activatable cell penetrating peptides that become effective only in the tumor microenvironment [110,111]. We have conceived a modular tool, dubbed HK2pep, constituted by four moieties: the active portion that mimics the N-terminal sequence of HK2 and detaches it from mitochondria; a polycation stretch required for crossing the plasma membrane, which is shielded by a polyanion sequence to avoid non-specific uptake in healthy cells; and a connector between these two charged domains that contains a cleavable consensus sequence for Matrix Metalloproteases Proteases 2 and 9 (MMP-2/9) that are often overexpressed in the milieu of many neoplasms, where they can detach the protective sequence and unleash the cell-penetrating HK2pep. This structure makes HK2pep suitable for intraperitoneal and intravenous administration and blood flow circulation, and indeed it turned out to be effective in inhibiting growth of colon and breast cancer cells allografted in nude mice without any evident off-target effect on healthy organs and tissues [92].

## 5. Conclusions

Mitochondrial HK2 modulates glycolysis and ROS levels, participates to Ca^2+^ signalling/homeostasis to control cell bioenergetics survival and to autophagy induction when nutrients are scarce. These variegated activities place HK2 at center stage in the unfolding process of growth and survival of neoplastic cells, contributing to their plasticity and adaptability in reacting to environmental changes.

These considerations make HK2 an interesting target for the development of novel antineoplastic approaches. Even if we need to go through more steps for reaching effective chemotherapeutics, the evolution of HK2-targeting peptides and/or the design of new molecules allosterically targeting HK2 is a promising perspective. HK2 targeting molecules could also synergize with traditional antitumor therapies, with the potential of erasing chemoresistant malignancies and limiting the minimal residual disease.

## Figures and Tables

**Figure 1 ijms-22-04716-f001:**
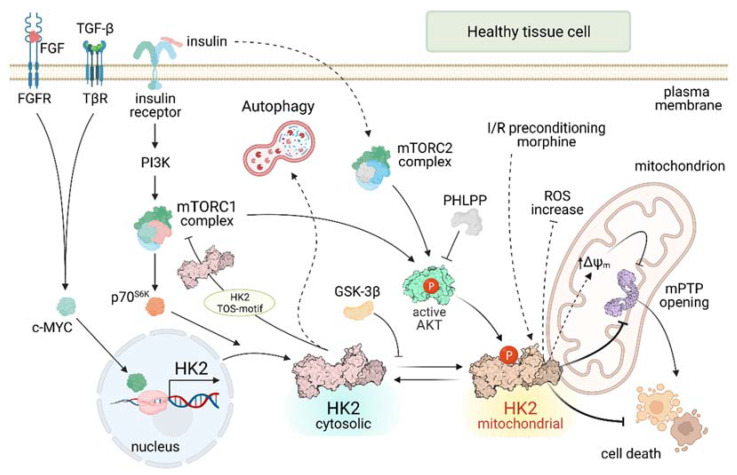
HK2 regulatory pathways in healthy cells. Continuous lines indicate direct effects; dashed lines indicate indirect effects. FGF: Fibroblast Growth Factor; FGFR: Fibroblast Growth Factor Receptor; TGF-β: Transforming Growth Factor β; TβR: Transforming Growth Factor β Receptor; PI3K: PhosphoInositide-3-Kinase; mTORC: mammalian Target of Rapamycin Complex; p70S6K: p70-S6 Kinase 1; GSK-3β Glycogen Synthase Kinase 3 β; PHLPP: PH domain and Leucine rich repeat Protein Phosphatase; ΔΨ_m_: mitochondrial membrane potential; mPTP: mitochondrial Permeability Transition Pore.

**Figure 2 ijms-22-04716-f002:**
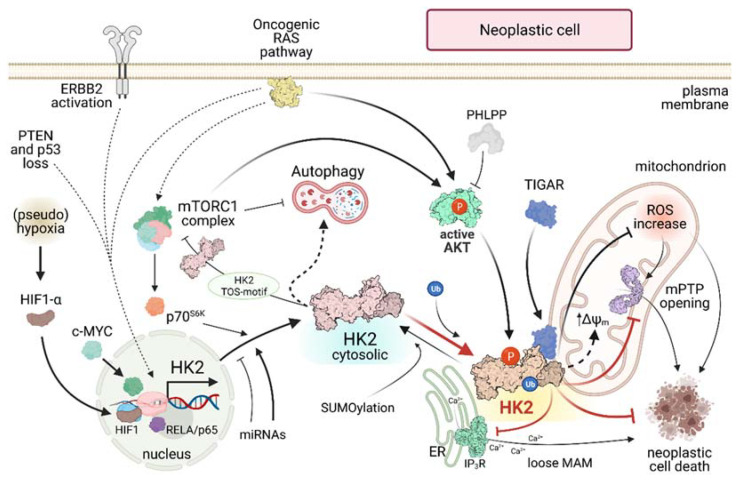
HK2 regulatory pathways in neoplastic cells. Continuous lines indicate direct effects; dashed lines indicate indirect effects. ERBB2: ERB-B2 Receptor Tyrosine Kinase 2; PTEN: Phosphatase and tensin homolog; p53: tumor suppressor p53; HIF: Hypoxia Inducible Factor; RELA/p65: nuclear factor NF-kappa-B p65 subunit; mTORC: mammalian Target Of Rapamycin Complex; p70S6K: p70-S6 Kinase 1; PHLPP: PH domain and Leucine rich repeat Protein Phosphatase; TIGAR: TP53 Induced Glycolysis Regulatory Phosphatase; ROS: Reactive Oxygen Species; ΔΨ_m_: mitochondrial membrane potential; mPTP: mitochondrial Permeability Transition Pore; ER Endoplasmic Reticulum; IP_3_R: Inositol-3-Phosphate Receptor; MAM: Mitochondria-Associated Membrane; P: Phosphorylation; Ub: Ubiquitination.

**Figure 3 ijms-22-04716-f003:**
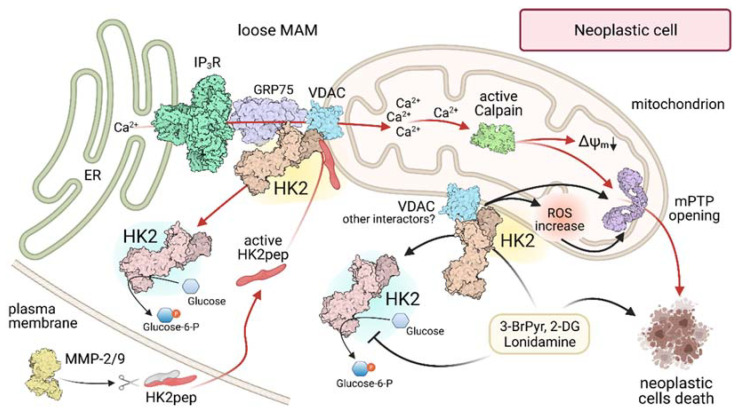
Modulation of HK 2 localization in MAMs of neoplastic cells. Black arrows indicate the action of 3-Br-Pyruvate (3-BrPyr), 2-DeoxyGlucose (2-DG) and Lonidamine; red arrows highlight the effects triggered by administration of a HK2-targeting peptide (HK2pep). MAM: Mitochondria-Associated Membrane; ER Endoplasmic Reticulum; IP_3_R: Inositol-3-Phosphate Receptor; GRP75: 75 kDa Glucose-Regulated Protein; VDAC: Voltage-Dependent Anion Channel; MMP-2/9: Matrix-MetalloProtease 2 or 9; ΔΨ_m_: mitochondrial membrane potential; mPTP: mitochondrial Permeability Transition Pore; ROS: Reactive Oxygen Species.

**Table 1 ijms-22-04716-t001:** Correlation of HK2 (over)expression with prognosis, tumor progression and treatment resistance in different type of human and murine neoplasms.

Tumor Type	HK2 Expression	Resistance	References
Hepatocellular carcinoma (HCC)	Poor prognosis; tumor progression	Sorafenib	[45,60,61,62,67]
Colorectal cancer	Poor prognosis; tumor progression		[54,60,61]
Glioblastoma	Poor prognosis; tumor progression	Radiotherapy	[55,66]
Pancreatic cancer	Poor prognosis	Gemcitabine	[56]
Cervical squamous cell carcinoma	Poor prognosis	Radiotherapy; 4-hydroxytamoxifen	[39,57]
Prostate Cancer	Poor prognosis; tumor progression		[58,63,64]
Head and neck squamous cell carcinoma	Poor prognosis		[59]
Gastric cancer	Poor prognosis		[60,61]
Lung cancer	Tumor progression		[15]
Breast cancer	Tumor progression	Paclitaxel	[15,38]
Diffuse large B-cell lymphoma	Tumor progression	R-CHOP regimen: Rituximab + cyclophosphamide, doxorubicin, vincristine and prednisone	[65,68]
Ovarian Cancer		Cisplatin	[40]

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
