# Peer review of "Hexokinase 2 in Cancer: A Prima Donna Playing Multiple Characters"

_ijms, 2021, doi:10.3390/ijms22094716_

Round 1

Reviewer 1 Report

The review article written by Ciscato F, et al., is highly informative and the text is well written. The information in the text includes recent advancement in the field, especially when there is a compelling evidence that mitochondrial-bound HK is acting as both a facilitator and gatekeeper of the malignant state in many cancers. The author has discussed a new modular tool, dubbed HK2pep usage that activates only in a tumor cell environment, where the death pathway is activated by displacing HK2 from OMM.  Overall, the review article advances the knowledge in the current field. The manuscript is suitable for publication. The author used sufficient reference.

Author Response

We are glad to hear that our contribution to this field have been appreciated and we thank the Reviewer for judging our manuscript suitable for publication.

Reviewer 2 Report

Hexokinase 2 in cancer: a prima donna playing multiple characters

Ciscato et al. in this review have detailed about the role of Hexokinase 2 (HK2) in tumor growth and proliferation. The authors have vividly explained how intra-cellular localization of HK2 promotes tumor growth and survival and inhibits apoptotic signaling. HK2 promotes metabolic rewiring in tumor cells leading towards aerobic glycolysis and away from oxidative phosphorylation. HK2 binds to OMM and inhibits pro-apoptotic signaling thereby providing protective effect to tumor cells. HK2 controls delicate Ca2+ flux from ER to mitochondria inhibition of which leads to lethal Ca2+ flux to mitochondria and cell apoptosis. Finally, they have briefly eluded to a novel peptide which could displace HK2 from the tumor cells without any side effect on normal organs or tissue.

PROS

The review article is well written and provides the reader with excellent overview of role of HK2 in tumor growth and proliferation.  The authors have excellently drafted the manuscript and provided relevant background including the role of HK2 in aerobic glycolysis and how its subcellular localization is affecting cells survival and growth. They have also eluded to many negative feedback loops which regulate HK2 signaling in case of nutrient deficiency. The review is very well written and provides a thorough summary of the role of HK2 in cancer.

CONS

Although the authors have briefly eluded to the fact there are no clinical trial ongoing with HK2 inhibitor currently, a simple search online provides you few examples of HK2 inhibitor which are currently being used in clinical trial to treat high grade gliomas such as Ketoconazole or Posaconazole. The authors need to modify this statement. A tabulated form explaining the reason for failure of different HK2 inhibitors and at what stage these therapies failed would make it much easier to read.

Author Response

We thank the Reviewer for positively evaluating our manuscript. Following her/his suggestion, we have carefully checked Hexokinase 2-associated clinical trials, adding in the text a part referring to Ketoconazole and Posaconazole as Hexokinase 2 inhibitors and as potential antineoplastic agents (highlighted in blue in the new version of the manuscript; lines 349-354 and 662-664). These are the only HK2-related compounds that have ever undergone any trial, albeit at present very preliminary, in cancer settings.